# Transfer-Free Analog and Digital Flexible Memristors Based on Boron Nitride Films

**DOI:** 10.3390/nano14040327

**Published:** 2024-02-07

**Authors:** Sibo Wang, Xiuhuan Liu, Han Yu, Xiaohang Liu, Jihong Zhao, Lixin Hou, Yanjun Gao, Zhanguo Chen

**Affiliations:** 1State Key Laboratory of Integrated Optoelectronics, College of Electronic Science and Engineering, Jilin University, Changchun 130012, China; wangsb21@mails.jlu.edu.cn (S.W.); hany23@mails.jlu.edu.cn (H.Y.); xiaohang21@mails.jlu.edu.cn (X.L.); zhaojihong@jlu.edu.cn (J.Z.); gaoyanjun6666@sina.com (Y.G.); 2College of Communication Engineering, Jilin University, Changchun 130012, China; xhliu@jlu.edu.cn; 3College of Information Technology, Jilin Agricultural University, Changchun 130118, China; houlixin_2000@126.com

**Keywords:** hexagonal boron nitride, memristor, artificial neural network, memory device

## Abstract

The traditional von Neumann architecture of computers, constrained by the inherent separation of processing and memory units, faces challenges, for instance, memory wall issue. Neuromorphic computing and in-memory computing offer promising paradigms to overcome the limitations of additional data movement and to enhance computational efficiency. In this work, transfer-free flexible memristors based on hexagonal boron nitride films were proposed for analog neuromorphic and digital memcomputing. Analog memristors were prepared; they exhibited synaptic behaviors, including paired-pulse facilitation and long-term potentiation/depression. The resistive switching mechanism of the analog memristors were investigated through transmission electron microscopy. Digital memristors were prepared by altering the electrode materials, and they exhibited reliable device performance, including a large on/off ratio (up to 10^6^), reproducible switching endurance (>100 cycles), non-volatile characteristic (>60 min), and effective operating under bending conditions (>100 times).

## 1. Introduction

The demand for computation and storage has increased rapidly with the development of the information age [1,2]. When the processor’s speed outpaces the rate at which data can be transferred to and from the memory system, the traditional von Neumann architecture encounters the memory wall problem, due to the physical limitations of the dynamic random-access memory (DRAM) and flash, facing challenges in further enhancing computational performance [3]. The memristor, as a two-terminal passive device with memory functionality, is considered a promising candidate to solve the problem in fields such as storge, artificial neural network computing, and so on. Since the successful manufacturing of the first memristor in 2008 [4], researchers have been paying increasing attention to memristor research. According to the primary applications of memristors, they can be categorized into two types: digital memristors and analog memristors [5]. Digital memristors exhibit a high switch ratio, fast transition speed between the high-resistance state (HRS) and low-resistance state (LRS), and stable voltage values for SET and RESET voltages. On the other hand, analog memristors, under the influence of applied bias, experience a gradual change in current without abrupt mutations. Their current variation is relatively smooth, and resistance changes with the applied scanning voltage, possessing the ability to memorize the current resistance state. Due to their distinct characteristics, these two types of memristors can be applied in different domains. Digital memristors are typically employed in storage and logical operations, making them suitable for using as storage units in systems such as computers [6,7]. Analog memristors, due to their synaptic-like characteristics, can be utilized to emulate synaptic plasticity, which makes them applicable in artificial neural networks (ANNs) and neuromorphic computing and simulation [8,9].

Flexible electronic devices, representing an emerging electronic technology, possess attributes that include being ultrathin, lightweight, flexible, and bendable. They have many applications in portable devices, wearable electronic products, and interfaces for human–computer interaction [10,11]. With the ongoing advancement of technology and the increasing demand for potential applications in the realm of next-generation flexible electronics, the advent of two-dimensional (2D) materials with high mechanical strength and hardness has opened new avenues for the advancement of flexible electronic devices. Notably, graphene, hexagonal boron nitride (hBN), and transition metal disulfide compounds (TMDCs) stand out as exceptional options in the pursuit of flexible electronic devices, due to their scalability and impressive mechanical flexibility [12,13,14,15,16,17]. However, due to the limitations of the physical properties of graphene and some TMDC materials, they are not suitable for preparing high-performance memristors. Among these materials, hBN exhibits a large bandgap of 5.97 eV, high in-plane thermal conductivity, and inherent flexibility [18,19,20,21]. These attributes contribute to a large on/off ratio, effective heat dissipation, and realizing flexibility for memristors. Therefore, it possesses significant potential as a high-quality dielectric layer for achieving flexible, non-volatile memristors [22,23].

The synthesis of 2D hBN can be achieved through top-down methodologies, such as mechanical and liquid-phase exfoliation [24,25]. Alternatively, it can be cultivated through bottom-up methodologies utilizing conventional techniques, such as low-pressure chemical vapor deposition (LPCVD) and magnetron sputtering [26,27]. In these growth methods, LPCVD is a widely researched material preparation technique. The fundamental principle of LPCVD involves introducing one or more precursors into a reaction chamber, allowing precursors to react or decompose on the substrate. This method has been used effectively in synthesizing 2D hBN and other ultrathin 2D materials [28]. Ammonia borane or borazine can be used as a precursor. Transition metal materials are used as substrates for synthesizing 2D hBN using the LPCVD method, owing to their catalytic and self-limiting growth effects on hBN [29,30,31]. These precursors are vaporized and introduced into the growth chamber containing the substrate, where they decompose on the substrate under a high temperature. The LPCVD process holds a lot of benefits, including controlled experimental parameters, large film deposition areas, and compatibility with traditional lithographic techniques, addressing many compatibility issues associated with the use of 2D materials in microelectronic devices.

Currently, there have been several studies on the resistive switching characteristics of the hBN films, confirming the feasibility of hBN memristors; even the monolayer hBN film can exhibit resistive switching characteristics [22,32,33]. Additionally, some researchers focus their attention on the resistive switching mechanisms of hBN memristors. If active materials such as silver or titanium are used for the one-side electrode, and a relative inert material is used for another side, the dominant mechanism in the device conforms to the electrochemical metallization theory; when a forward voltage is applied to the active metal electrode side, these active metals will be oxidized and will move to the inert electrode side and reduce, thereby forming conductive filaments in the hBN memristor, causing the device to transform the resistance state from HRS to LRS [34,35,36]. When inert metal materials are used for the electrodes at both ends of the memristor, the memristive phenomenon can be attributed to the influence of defects in the film [37]. Additionally, when metal electrodes form conductive filaments, memristive devices may exhibit resistive switching behaviors through a thermochemical mechanism. If the metal constituting the conductive filaments has a lower melting point and imposes a schemed compliance current, the device exhibits a resistive switching mechanism involving Joule heating-induced dissolution; the conductive filaments are thermally melted, thereby changing the device state from LRS to HRS [38]. The resistance switching characteristics of hBN memristors can also be combined with multiple conductive mechanisms [39]. Herein, we showed the fabrication of transfer-free, flexible hBN memristors and modulated the characteristics of the devices by varying the material of the top electrode, achieving analog memristors with biological synaptic features and digital memristors with a large on/off ratio. The resistive switching characteristics of hBN memristors were investigated. The effects of two different electrode materials, Ag and Al, as top electrodes for a metal/hBN/Cu foil structure memristor were explored. The analog characteristic was suggested to be associated with the multi-filament configuration and rupture dominated by ion migration under the influence of an external electric field. The digital characteristic, on the other hand, was attributed to filament rupture induced by the Joule heating effect.

## 2. Experiments

### 2.1. Fabrication and Transfer Process of the hBN Films

The hBN film was synthesized on copper foil with a thickness of 25 μm using the LPCVD method. A 5 mg quantity of borazane in a chamber with an individually heating belt was used as the source supply. The Cu substrate was annealed for 60 min at 100 Pa and 1273 K in a mixture of hydrogen and argon gases, with flow rates of 50 sccm for both gases, before the growth process. During the growth process, the carrier gases, consisting of high-purity argon and hydrogen, were also set at flow rates of 50 sccm, individually. The overall gas flow was 100 sccm while maintaining a total pressure of 100 Pa and a temperature of 1273 K. The growth process persisted for a duration of 5 min. After growth, the borazane valve was closed, and the furnace underwent cooling down to room temperature under an argon atmosphere with a flow of 100 sccm.

For further characterization, the films were transferred to the SiO_2_ (300 nm)/Si substrate using a wet transfer method. The preparation processes were as follows: A total of 3 g of polymethyl methacrylate (PMMA) powder was dissolved in 50 mL of cyclopentanone and stirred for 8 h with magnetic flux to prepare the PMMA solution. Then, the PMMA solution was spin-coated onto the grown hBN sample at a low rotation speed of 500 revolutions per second (rpm) for 6 s, followed by a high rotation speed of 2000 rpm for 30 s. Subsequently, the PMMA-coated sample was cured on a hotplate at 353 K for 10 min. The sample was then immersed in a prepared 5 mol/L ferric chloride solution for 5 h. Then, the sample was placed in deionized water to remove any remaining copper after corrosion. The sample was then transferred onto the SiO_2_/Si substrate and placed into acetone for 5 h to dissolve the PMMA. To ensure complete removal, the sample was then subjected to high-temperature annealing in a hydrogen atmosphere at 1073 K for 1 h using a hydrogen flow rate of 50 sccm at atmospheric pressure, ensuring the absence of residual PMMA.

### 2.2. Fabrication of Devices

The copper foil on which the hBN film was grown was utilized as the bottom electrode (BE) for the device, while the top electrode (TE) was prepared by electron beam evaporation. At the end, TEs were deposited onto the hBN film through a metal mask with pores (each with a radius of 1 mm). After the completion of metal electrode preparation, these samples were placed in an annealing furnace to anneal the electrodes under ambient pressure. The annealing time was set to 10 min at a temperature of 673 K, with the introduction of 50 sccm argon and 50 sccm hydrogen. Hydrogen was used to prevent the oxidation of the metal electrodes at high temperatures. After annealing, the samples were cooled naturally to room temperature. Through this method, we obtained Ag/hBN/Cu foil memristors, with a 200 nm thickness for Ag. The same procedure was employed to fabricate Au/Al electrodes, resulting in Au/Al/hBN/Cu foil memristors, with a 100 nm thickness for both Au and Al.

### 2.3. Characterizations of hBN Film and Memristor Devices

Atomic force microscopy (AFM, BRUKER ICON-PT, Billerica, MA, USA) was used to characterize the surface of the fabricated hBN film. Raman scattering spectroscopy (Linkam Scientific, LabRAM HR evolution, Salfords, UK) was carried out to determine the crystalline phase and bonding configurations of hBN films. Chemical states of the hBN films were analyzed using an X-ray photoelectron spectrometer (XPS, Thermo Scientific: ESCALAB 250Xi, Waltham, MA, USA) with a focused monochromatic Al Ka X-ray source. To conduct electrical characterization, the test sample was placed on a probe device, and flexible Pt probes with a diameter of 1 micron were used. The probes were in direct contact with the top and bottom electrodes for testing purposes. The current–voltage (I–V) measurements were performed using source meters (Keithley 2410 for on/off characteristics and keysight B2902A for synaptic properties), with the positive bias defined as when the current was flowing from the TE to the Cu foil. Morphologies of the devices were investigated using cross-sectional transmission electron microscopes (TEM, JEOL JEM-F200, Tokyo, Japan). The cross-sectional TEM sample was prepared by using a focused ion beam (FIB, Thermo Fisher Scientific HELIOS 5 CX DualBeam, Waltham, MA, USA), which can cut thin lamellas from the devices, and then was placed on a copper grid for TEM inspection. For further investigation, the chemical composition of the conducting filament was characterized using an energy-dispersive spectrometer (EDS) in scanning TEM (STEM) mode.

## 3. Results and Discussion

### 3.1. Characterizations of the hBN Film Grown on Copper Foil

The hBN sample was characterized by XPS. As shown in Figure 1a, the peak positions of the B1s orbital electrons were at 189.98 eV, the N1s orbital electrons were at 397.58 eV, and the B/N ratio was 1.10, which are in good agreement with the XPS peak positions of hBN reported in the literature [40]. These binding energies are directly related to the hexagonal B-N bonding, implying the hexagonal phase of the BN film. The crystalline phase of the hBN film was investigated by Raman spectra between 1200 cm^−1^ and 1600 cm^−1^, as shown in Figure 1b, featuring a characteristic hBN peak (E_2g_) at 1371.96 cm^−^¹, indicating the film of sp^2^-hBN [41]. The film was transferred to the SiO_2_/Si substrate, characterized using an optical microscope and AFM, as shown in Figure 1c,d. The AFM characterization results obtained at the red line position showed that the film thickness was determined by measuring the steps on the surface of the SiO_2_/Si substrate and the hBN film. The measurements at the film edge revealed a thickness of 45.3 nm all around.

### 3.2. Characterizations of hBN-Based Memristor Devices

hBN memristors have the ability to modulate resistance states and hold significant potential as electronic synapses in in-memory computing. We used electron beam evaporation to prepare circular top electrodes (radius = 1 mm) on the grown hBN film and obtained Ag/hBN/Cu foil-structured memristors on a polyethylene terephthalate (PET) substrate, with Cu foil as the BE. One of the common methods for preparing 2D hBN devices includes growing hBN at high temperatures on a catalytic metal substrate surface by CVD, transferring the prepared hBN to the target substrate, and fabricating the device using lithographic processes. The transfer process may lead to wrinkles, damage, and the contamination of hBN. Using Cu foil, which has grown hBN, as the bottom electrode eliminates the need for the transfer process. The transfer-free method simplifies the device fabrication process, reducing potential damage and contamination to the film during the transfer process of 2D materials. The structure of the hBN memristor was designed as shown in Figure 2a,b, employing a vertical configuration of metal–insulator–metal (MIM), a common structure in memristor research. Figure 2c illustrates the forming process and the typical bipolar resistive switching characteristics curve of the Ag/hBN/Cu foil memristor. Before the forming process, the resistance value of the Ag/hBN/Cu foil device was around 10^7^ Ω. After the forming process, the resistance value of the device was reduced to 10^2^ Ω all around, and the device exhibited bipolar resistive switching (RS) behaviors. The blue line of Figure 2d represents the 100th I–V curve. Even after 100 cycles of endurance testing, the device still maintained bipolar RS behavior, as shown in Figure 2d.

Figure 3a depicts the electrical characteristics of the hBN memristor under ten consecutive positive voltage scans. During the positive bias voltage scans, the gradually increasing current illustrates the significant potential of the hBN memristor with resistive modulation; the resistance value of the device decreased from 253 Ω to 169 Ω, which indicates its substantial promise as an electronic synapse in memory computing. Synaptic plasticity is considered to provide the brain with the capability to learn and process information [42,43]. Long-term potentiation/depression (LTP/LTD) constitutes the fundamental synaptic plasticity in analog neuromorphic computing in ANNs, and it can be emulated by applying a series of pulses to the electronic synapse, as demonstrated in Figure 3b, exhibiting LTP/LTD characteristics during the process. The current plateau is caused by the CC. LTP enables experience-based learning to occur over minutes or longer time scales and forms the biological foundation of the Hebbian learning rule (i.e., weight updates) in neural morphological systems. LTD refers to a persistent decrease in the effectiveness of neuronal synapses that occurs hours or more after exposure to a prolonged patterned stimulus.

Through the application of pulse voltage stress (PVS) sequences with varying amplitudes, durations, and intervals (matched to actual working conditions), the dynamic responses of hBN synapses were investigated, with simultaneous monitoring of the current between them. The synapses were exposed to continuous PVS sequences while applying voltage through I–V scans, with a pause between each sequence. The voltage upswing was set at 0.4 V, and during the voltage application, the current through the synapse gradually increased, as depicted in Figure 3c. This behavior was applied to simulate paired-pulse facilitation (PPF) in biological synapses.

In order to investigate the resistive switching mechanism, we examined and studied the Ag/hBN/Cu memristor through TEM. Prior to TEM testing, the device underwent a forming process, and TEM samples were prepared by FIB. Figure 4a shows the TEM image of the device structure without the forming process. At this moment, no conductive pathways were formed internally in the memristor. During the forming process, the large size of our device’s TE resulted in the formation of multi-filaments, some of which were already conducting or about to conduct, as depicted in Figure 4b. To further explore which material formed the conductive filaments inside the memristor, we performed EDS tests on the device. The EDS X-ray maps, as shown in Figure 4c–e, show one of the conductive paths (red circle) in Figure 4b, indicating that the conductive paths in the Ag/hBN/Cu foil memristors were primarily composed of Ag, which can be attributed to the different electrochemical activities between Ag and Cu foil; the Ag filament nucleation required a lower electric field, compared to Cu [44]. Due to the presence of numerous defects in the prepared film, conductive filaments were more prone to forming inside the film [39]. Based on the above analysis, the reason for the device’s memory characteristics with synaptic behavior can be attributed to the modulation of multi-filaments composed of Ag. As schematically illustrated in Figure 4f, when a single reverse voltage was applied, some of the conductive pathways were disrupted. However, due to the large top electrode, many conductive paths formed inside the device, and some conductive paths were still not disconnected. When more pulses were applied, more conductive pathways were connected or disconnected, resulting in the device exhibiting characteristics similar to an analog device.

To investigate the influence of electrode materials on resistive switching characteristics, we employed a similar fabrication method to prepare Au/Al/hBN/Cu foil-structured memristors. The Au layer served as a protective layer to prevent the oxidation of aluminum in ambient air. Figure 5a illustrates the typical bipolar resistive switching behavior of the Au/Al/hBN/Cu foil memristor as the voltage sweeps from 0 to 1 V. When a SET voltage was applied to the device, the memristor transitioned from a high-resistance state to a low-resistance state. Conversely, when a RESET voltage was applied, the device returned from a high-current state to a low-current state, corresponding to the RESET process. The on/off ratio of the memristor was greater than 10^6^. Before testing, the initial resistance of the Au/Al/hBN/Cu foil memristor was 3.13 × 10^7^ Ω. Under the condition of a read voltage of 0.1 V, the memristor underwent cyclic testing, and even after 100 cycles, the device still exhibited clear resistive switching characteristics, as shown in Figure 5b. The SET and RESET voltages were counted from the durability test, and the distributions of SET and RESET voltages are shown in Figure 5c. Further performance testing results indicated that the device still exhibited resistive switching characteristics after 60 min, as Figure 5d shows, which indicates that the memristor is a non-volatile memory. The resistive switching device of the Au/Al/hBN/Cu foil had a large switching ratio (up to 10^6^). We believe this can be attributed to the magnitude of CC and the lower melting point of Al, compared to Ag. If the CC is set at a smaller value, the device may exhibit similar characteristics as the Ag/hBN/Cu foil memristor. However, with a designed CC and voltage scanning amplitude, bipolar resistive characteristics will exist in the Au/Al/hBN/Cu foil memristor. If an appropriate CC is not set, the device may exhibit unipolar or nonpolar characteristics [38] because the device is subject to several resistance mechanisms working together internally. Under the combined influence of electrochemical metallization and Joule heating, the application of a reverse voltage, according to the theory of electrochemical metallization, gradually leads to thinner conductive filaments, making them more susceptible to rupture by Joule heating. Under positive bias, with a designed compliance current, the thicker conductive pathways are less prone to being disconnected by Joule heating. Under the combined influence of reverse bias and Joule heating, the conductive paths that have been formed are entirely disconnected, resulting in a significant on/off ratio. The transition in resistive states of hBN memristors induced by Joule heating has been reported in other literature [38]. Such a resistive switching mechanism has been similarly reported in resistive switching devices with metal oxides as the dielectric [5].

To investigate the flexibility and durability of the hBN memristors, folding–relaxation tests are crucial for flexible devices. We tested the durabilities of two different devices with distinct structures that we prepared. Figure 6a illustrates the tolerance test of the hBN memristor with an Ag/hBN/Cu structure. Durability tests of LTP/LTD were conducted with pulses to validate reliable computing capability based on weight updates in the flat state. After 100 repeated folding–relaxation cycles with a radius of 10 mm, as shown in Figure 6a, the proposed memristor continued to exhibit continuous LTP/LTD behavior without significant degradation, indicating high durability, even under mechanical strain. Similarly, we also conducted tests on the hBN memristor with an Au/Al/hBN/Cu structure. We explored the bending reliability of the proposed hBN resistive switching device. After 100 bending tests, the memristor still exhibited evident resistive switching characteristics, as Figure 6b demonstrates, indicating excellent mechanical robustness. The on/off ratio of the device showed little change, suggesting excellent bending stability. This indicates that the Au/Al/hBN/Cu foil and Ag/hBN/Cu foil devices have outstanding mechanical robustness, making these devices suitable for flexible, non-volatile memory applications.

## 4. Conclusions

In summary, transfer-free flexible hBN memristors with bipolar resistive switching characteristics for memory and computing were presented. The LPCVD technique was utilized to grow hBN films on 25 μm thick copper foil, and two prototypes of hBN memristor devices with different characteristics were prepared. Digital and analog memristors were achieved by changing the top electrode material. The analog memristor demonstrated synaptic plasticity and neural behaviors, with features like LTP/LTD and PPF. Multiple conductive paths inside the analog memristor were observed through TEM, and the resistive switching mechanism was explained. The fabricated digital memristor had a high switch ratio (>10^6^), indicating its potential for applications in the storage field. The device maintained stable resistive switching characteristics, even after 100 bending cycles. The research results presented in this paper provide insights for the development of next-generation wearable memory computing systems capable of implementing digital and analog neuromorphic computing functions.

## Figures and Tables

**Figure 1 nanomaterials-14-00327-f001:**
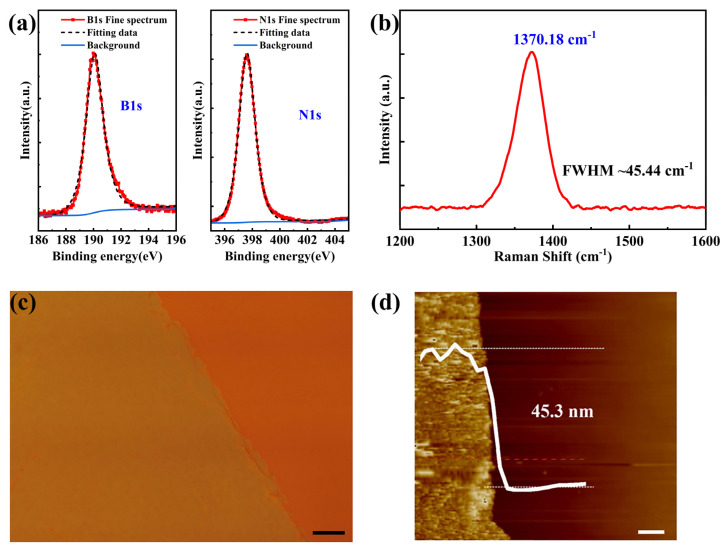
XPS, Raman, optical microscopy, and AFM characterization results of the films. (**a**) XPS spectra of B1s and N1s. (**b**) Raman spectra of the hBN film. (**c**) Optical microscopy image of the hBN film transferred onto SiO_2_/Si substrates. (Scale bar = 5μm). (**d**) AFM images and height profiles of the transferred hBN film on SiO_2_/Si substrates. (scale bar = 1 μm).

**Figure 2 nanomaterials-14-00327-f002:**
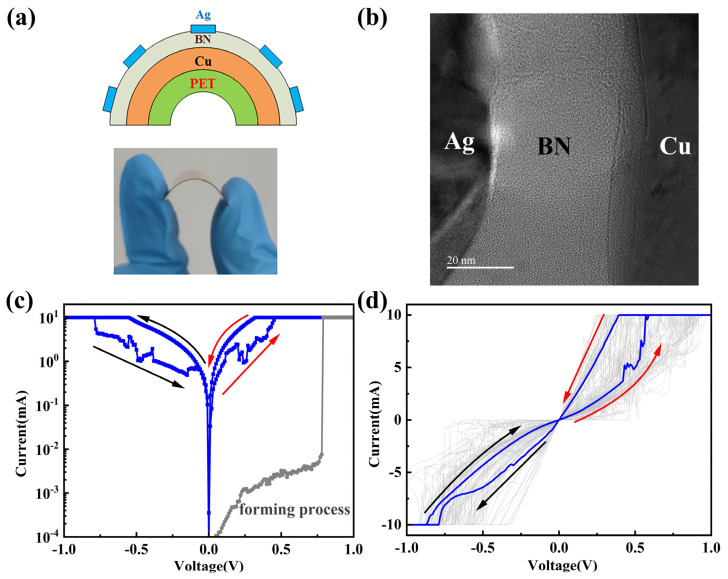
Schematic of the Ag/hBN/Cu/PET and testing of memristive characteristics. (**a**) Schematic of the Ag/hBN/Cu foil on the PET substrate device. (**b**) Cross-section TEM image of the Ag/hBN/Cu foil memory cell. (**c**) Typical bipolar characteristic and forming process of the memristor. (**d**) The I–V characteristics of 100 consecutive sweeping cycles, exhibiting bipolar switching characteristics. The blue line represents the 100th I–V curve. The compliance current (CC) is 10 mA.

**Figure 3 nanomaterials-14-00327-f003:**
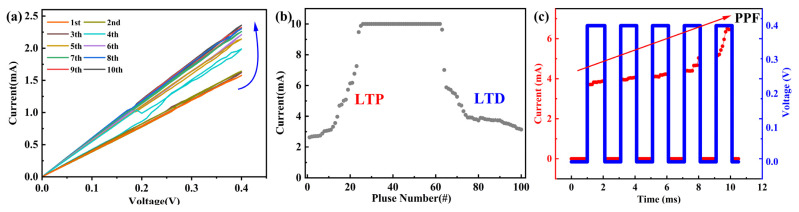
The dynamic responses of Ag/hBN/Cu foil memristor as synapses. (**a**) The current of the electronic synapse under 10-times positive voltage sweeps. (**b**) LTP and LTD stimulated by 50 positive and negative pulses (0.4 V/−0.4 V, CC = 10 mA); each pulse lasts for 0.1 ms. (**c**) The STP of the synaptic device is shown experimentally by PPF.

**Figure 4 nanomaterials-14-00327-f004:**
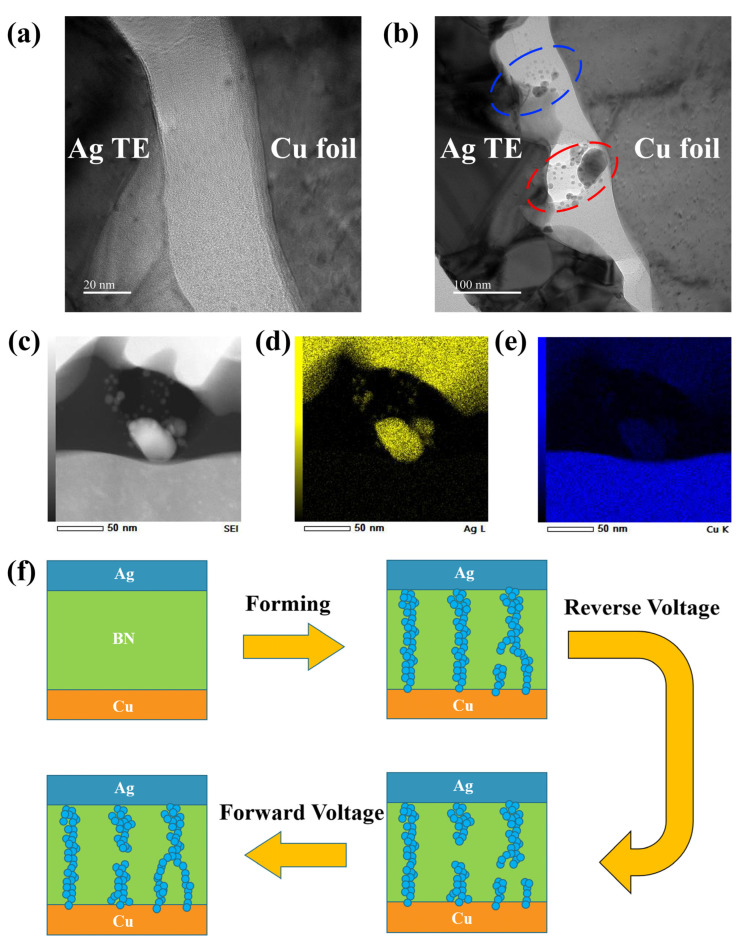
The TEM image of the device before (**a**) and after (**b**) the forming process. (**c**) Image of conductive pathways. The EDS mapping of Ag (**d**) and Cu (**e**). (**f**) Schematic diagram of the resistance switching mechanism of multiple conductive filaments for the Ag/hBN/Cu foil memristor.

**Figure 5 nanomaterials-14-00327-f005:**
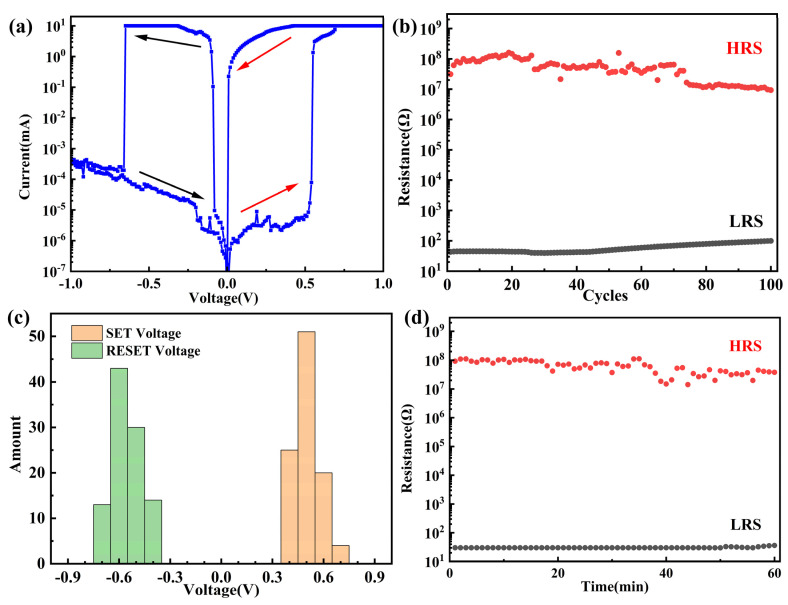
The memristive characteristics tests of the Au/Al/hBN/Cu foil memristor. (**a**) Typical bipolar characteristic and forming process of the memristor. (CC = 10 mA). (**b**) The endurance of the memristor under SET/RESET voltage pulses over 100 cycles. (**c**) SET and RESET voltage distributions of 100 cycles. (**d**) Retention time of the device.

**Figure 6 nanomaterials-14-00327-f006:**
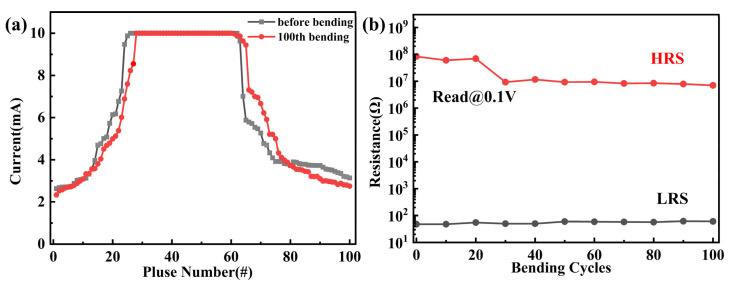
Folding–relaxation tests of two kinds of hBN memristors. (**a**) After 100 bending cycles by 50 positive and negative pulses of the Ag/hBN/Cu-structured memristor, each pulse lasting 0.1 ms (0.4 V/−0.4 V), LTP and LTD were stimulated. (**b**) The Au/Al/hBN/Cu structure device distributions of HRS and LRS after 100 bending cycles.

## Data Availability

Data are contained within the article.

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
