# Peer review of "Transfer-Free Analog and Digital Flexible Memristors Based on Boron Nitride Films"

_nanomaterials, 2024, doi:10.3390/nano14040327_

Round 1

Reviewer 1 Report

Comments and Suggestions for Authors

The paper is dedicated to very important fiedl. Elements for neuromorphing computing are of great interest.

But this paper should be corrected a lot before being considered for publication.

The quality of graphical material is very pure.

Where are the fit of XPS and detailed description of peaks?

Diffraction from TEM is highly recommented to be added.

What about grafical abstract?

Comments on the Quality of English Language

English is OK.

Reviewer 2 Report

Comments and Suggestions for Authors

The manuscript presents hexagonal boron-nitrite based flexible resistive switching devices. The work seems relevant however, there are some questions that need to be addressed before it could be published:

1. The authors state that “the research on using hexagonal boron nitride as the dielectric material of memristor to prepare the device with flexibility is still in its infancy, and there are few literature reports on its resistive switching mechanism.” However, there are 85 results on Scopus for “hexagonal AND boron AND nitride AND resistive AND switching”, which shows that it is a highly studied material. Although not directly for flexible applications, as the resistive switching mechanism should be comparable, this work must be put in context with the previous literature.

2. The devices fabricated are quite large (1 mm in radius) and thick (45.3, 100 and 200 nm). Could the authors comment on the influence of these scales on the obtained results? Would the results be different for smaller sizes? Furthermore, the thickness of the copper foil should be specified.

3. Some experimental details are missing. Namely the process used to transfer the film to the SiO2/Si substrate. In Fig. 1(d) it is shown a thickness of 45.3 nm for the transferred hBN, however the film might have not been completely transferred, depending on the process, and therefore the original thickness (on the device), might be higher.

4. Also, the equipment used for the electrical measurements is not mentioned, as well as how the bottom and top electrodes are contacted, and which one is the reference.

5. What does it mean “obtained Ag/hBN/Cu foil structured memristors on a polyethylene terephthalate (PET) substrate”? Was he copper foil fixed on PET somehow?

6. Figs. 2 (c) and (d) should have the same units (it is A and mA). The direction of the cycle must be described and/or marked in the figure and the resistance values mentioned. Why is there no statistical analysis of voltage and resistance evolution shown for Ag (Fig.2), as it is the case of Al (Fig.5)?

7. The duration of the pulses in Fig. 3(b) is not mentioned.

8. Did the authors have some care in sample preparation for TEM to find the exact place with the “filaments”? In such a big area (1 mm radius), how do they know if these are indeed filaments responsible for resistive switching behavior or some fixed defects?

9. The schematic in Fig. 4(f) is not correct, as it shows a complete filament after Reset and this would imply a low resistance state.

10. It is said that “The resistive switching device of Au/Al/hBN/Cu foil has a large switching ratio, and the differentiation between high and low resistance states is significant. We believe this is attributed to the lower melting point of Al compared to Ag.”. What does “significant” mean? It needs to be quantified. Also, what is the initial resistance in both cases? If Joule heating plays a role, should the behavior be unipolar?

11. For clarity, the captions of all Figs should specify if Ag or Al electrode is being studied.

12. The authors call digital (instead of analog) memristor to the Al/hBN device, however showing LTP and LTD. It is confusing to call something digital and then show that it has more than two resistance states.

13. The quality of the images must be improved.

Comments on the Quality of English Language

Some minor English corrections are needed. For example, in the Abstract, the sentence “Analogue memristors of the synaptic behaviors are demonstrated” sounds odd.

Reviewer 3 Report

Comments and Suggestions for Authors

In this paper, the authors present a study on transfer-free flexible hBN memristors with bipolar resistive switching characteristics for memory and computing applications. By changing the top electrode material, from Ag to Al, the authors reported data on both digital and analog memristors, and especially investigated devices’ current-voltage characteristics and the associated low resistance states (LRS) and high resistance states (HRS), and their immunity to different stresses e.g bending cycles or set/reset pulses. The authors demonstrate that the “digital memristor” has a high switch ratio (>10^6) and that it maintains stable resistive switching properties after 100 bending cycles. This work has potential for the development of next-generation wearable systems that are capable of implementing both digital and analog neuromorphic computing functions. This work is interesting and potentially useful, but there are some issues that need to be addressed before this manuscript can be reconsidered for publication.

(1) On page 2, section 2.2, the authors state that Ag electrode is 200 nm thick, while Al electrode is 100 nm thick. Please briefly comment if you believe that different thickness has impact on metal atom migration and formation of conducting filaments, and consequently on the performance of memristors.

(2) On page 3, section 3.1, the hBN thickness is measured to be 45 nm – how flexible is a layer of this thickness? Can you estimate upper thickness that would make such layer inflexible, or brittle, i.e. inapplicable for flexible hBN memristor?

(3) Concerning explanations about filament creation on page 6, I suggest the authors to also comment that the stochastic nature of the filament creation and destruction is also dominantly responsible for memristor variability reported in other figures.

(4) Regarding the first analyzed device, Ag/hBN/Cu memristor in Fig. 2, please provide analysis of LRS and HRS values (cf. Fig.3a), and comment on device variability reported in Fig. 2d in terms of feasibility of these devices for practical applications.

(5) Regarding Fig.3b – what causes the current plateau? Is this the current-compliance limit in the measurement system, or something else?

(6) On page 4, first part of section 3.2, and on page 5, first part of the first paragraph, there is a lot of general knowledge. I suggest shortening these parts and providing references to the interested readers.

(7) Figures are generally of low quality or low resolution; some of them are barely readable. This must be improved to enable further review of data.

(8) On page 7, the authors claim that under the combined influence of reverse bias and Joule heating, the conductive paths that have been formed are entirely disconnected, resulting in a significant on/off ratio. These statements need more elaboration and/or support from references, as it is not clear that Joule heating is indeed responsible for the destruction of filaments in hBN memristors.

Comments on the Quality of English Language

Minor editing of English language required

Round 2

Reviewer 2 Report

Comments and Suggestions for Authors

The authors answered most of my points, however I must insist on the following ones:

6. Figs. 2 (c) and (d) should have the same units (it is A and mA). The direction of the cycle must be described and/or marked in the figure and the resistance values mentioned. Why is there no statistical analysis of voltage and resistance evolution shown for Ag (Fig.2), as it is the case of Al (Fig.5)?

>>The questions regarding the statistical analysis was not assessed.

7. The duration of the pulses in Fig. 3(b) is not mentioned.

Response 7:Thank you very much for the suggestion.In response to your comment, the duration of each pulse has been added to the description in Figure. 3(b).

(Lines from 227to 230of page 6)“Figure 3. The dynamic response of Ag/hBN/Cu foil memristor as synapses. (a) The current of the electronic synapse under 10 times positive voltage sweeps. (b) LTP and LTD stimulated by 50 positive and negative pulses (0.4 V/-0.4 V),each pulse lasts for 0.1 ms. (c) The STP of the synaptic device is shown experimentally by PPF.”

>>How can you apply and read with a 0.1 ms pulse using a using a “source meter (Keithley 2410)” (Response 4)? As far as I know it, this equipment resolution does not allow it.

13. The quality of the images must be improved.

>>The figures still look “blurry”, without high definition.

Comments on the Quality of English Language

The word "transferation" (p.3 line 112) must be changed and “memristors with biological synaptic features and digital memristors with large on/off ratio” should be “off/on ratio”.

Reviewer 3 Report

Comments and Suggestions for Authors

All comments have been addressed adequately.

Comments on the Quality of English Language

Minor improvements needed, can be done during proofing.
